# Enhancing Physics-Informed Neural Networks Through Feature Engineering

## Abstract

Recent advances in Physics-Informed Neural Networks (PINNs) have deployed fully-connected multi-layer deep learning architectures to solve partial differential equations (PDEs). Such architectures, however, struggle to reduce prediction error below $O(10^{-5})$, even with substantial network sizes and prolonged training periods. Methods that reduce error further, to $O(10^{-7})$, generally come with high computational costs. This paper introduces a **S**ingle-layered **A**rchitecture with **F**eature **E**ngineering (SAFE-NET) that reduces the error by orders of magnitude using far fewer parameters, challenging the prevailing belief that modern PINNs are effectively learning features in these scientific applications. Our strategy is to return to basic ideas in machine learning: with random Fourier features, a simplified single-layer network architecture, and an effective optimizer we call $(\text{Adam} + \text{L-BFGS})^2$, SAFE-NET accelerates training and improves accuracy by reducing the number of parameters and improving the conditioning of the PINN optimization problem. Our numerical results reveal that SAFE-NET converges faster, matches or generally exceeds the performance of more complex optimizers and multi-layer networks, improves problem conditioning throughout training, and works robustly across many problem settings. On average, SAFE-NET uses an order of magnitude fewer parameters than a conventional PINN setup to reach comparable results in $20\%$ as many epochs, each of which is $50\%$ faster. Our findings suggest that conventional deep learning architectures for PINNs waste their representational capacity, failing to learn simple but effective features.

## 1 Introduction

Partial Differential Equations (PDEs) form the bedrock of numerous scientific and engineering fields, but their solution poses fundamental difficulties. Very few PDEs possess analytic solutions, forcing scientists and engineers to rely on numerical methods, which become prohibitively expensive for non-linear and high-dimensional problems. In the past several years, approaches based on machine learning have been developed that have the potential to ameliorate this problem. Amongst the many approaches, one of the most popular is the physics-informed neural network (PINN) (Raissi et al., 2019; Karniadakis et al., 2021; Cuomo et al., 2022), which has the potential to solve both forward and inverse problems involving PDEs. PINNs parameterize the solution to a PDE with a neural network, and are often fit by minimizing a least-squares loss involving the PDE residual, boundary condition(s), and initial condition(s). As PINNs do not require a mesh, they have the potential to efficiently solve high-dimensional and non-linear problems that require massive computing power to solve with classical numerical methods.

Unfortunately, recent works Krishnapriyan et al. (2021); Rathore et al. (2024) have shown that PINNs are extremely difficult to train. The main source of the diffiuclty being the ill-conditioning induced by the differential operator appearing in the PINNs loss. The ill-conditioning makes it very difficult for popular optimizers such as Adam (Kingma & Ba, 2014) and SGD (Bottou, 2010) to make the loss small enough so as to yield a useful model.

To address this issue, Rathore et al. (2024) proposed a custom optimizer, NysNewton-CG (NNCG), which, when combined with Adam and L-BFGS, significantly improves the optimization of the PINNs loss function. This combination achieved error reductions of one to two orders of magnitude compared to using Adam and L-BFGS alone.

Despite its effectiveness, NNCG has two significant shortcomings: (1) it is computationally expensive and (2) it is difficult to tune. For the wave equation, Rathore et al. (2024) reports that a step of NNCG is over $10\times$ then a step of L-BFGS. It is partly because of this that they recommend making as much progress as possible with Adam and L-BFGS before switching to NNCG. NNCG also has multiple hyperparameters, that are not easily set apriori, which means an expensive grid-search may be required.

In this paper, we ask whether or not a simpler more computationally friendly solution is possible to address the challenges in training PINNs? We argue that the primary issue facing PINNs is that the input representation is inadequate for the network to adequately learn the solution. We show that by incorporating an informative set of features that better capture the inductive bias of the PDE, we can exceed the performance of more complex optimizers and multi-layer networks. In particular, our proposed method SAFE-NET, combines two-layer neural network with feature augmentation to yield a method that is not only faster and computationally less expensive but also more generalizable to a wider range of more complex problems.

### 1.1 CONTRIBUTIONS

- SAFE-NET challenges the prevailing belief that deeper networks are necessary for better learning features in scientific applications, showing that substantial reductions in error can be achieved faster with significantly fewer parameters. SAFE-NET utilizes on average only about 600 parameters—far fewer than the approximately 5300 parameters typically used in a conventional four-layer PINN. This drastic reduction in parameter count leads to faster epoch times and requires less than 20% of the epochs typically needed by traditional models to achieve comparable results.

- By utilizing the $(\text{Adam} + \text{L-BFGS})^2$ optimizer, SAFE-NET not only accelerates training but also enhances the overall accuracy, matching or exceeding the performance of more complex optimizers and multi-layer networks on several problems.

- SAFE-NET improves the spectral density of the loss landscape both early in training and after training, indicating better initialization and enhanced conditioning of the problem with SAFE-NET, which is critical for the successful training of PINNs.

## 2 PROBLEM SETUP

An overview of PINNs and the PINN loss function is provided in Appendix A. We conduct a series of experiments aimed at optimizing the PINN loss function for a range of PDE problems including wave, convection, heat, Burgers, and reaction equations. These PDEs have been rigorously studied in previous works, particularly in the context of training difficulties associated with PINNs. The coefficient settings are detailed in Appendix A.

Our experiments assess the performance of these optimizers: Adam, L-BFGS, a combination of Adam and L-BFGS (Adam+L-BFGS), and a custom hybrid optimizer referred to as $(\text{Adam} + \text{L-BFGS})^2$. For Adam, the learning rate is tuned via a grid search over the set $\{10^{-5}, 10^{-4}, 10^{-3}, 10^{-2}, 10^{-1}\}$. For L-BFGS, we use a default learning rate of 1.0, a memory size of 100, and employ a strong Wolfe line search. The combined Adam+L-BFGS and the $(\text{Adam} + \text{L-BFGS})^2$ setups involve tuning the Adam learning rate as previously mentioned, with adjustments in the switching sequence from Adam to L-BFGS to optimize results. For the first set of experiments, each optimizer is run for 20,000 epochs, and in the second set, the duration is extended to 40,000 epochs, switching from L-BFGS to Adam whenever a stall in L-BFGS is detected. Throughout this study, we compare our proposed architecture, SAFE-NET—simply a single-layer network—to a traditional PINN model, which is a multi-layer perceptron (MLP)-with three hidden layers-and tanh activations. Network initialization is performed using the Xavier normal initialization method with all biases set to zero. Each experimental configuration, encompassing combinations of PDEs, optimizers, and network structures, is replicated across seven random seeds to ensure robustness. The computational domain for each problem is discretized using 10,000 residual points randomly sampled from a $255 \times 100$ grid, with 257 equally spaced points for initial conditions and 101 equally spaced points for each boundary condition.

In each setup, the discrepancy between the predicted solution and the ground truth is evaluated using the $\ell_2$ relative error (L2RE), a standard metric in the PINN literature. Given the PINN prediction $\mathbf{y} = (y_i)_{i=1}^n$ and the ground truth $\mathbf{y}' = (y_i')_{i=1}^n$, the L2RE is defined as:

$$L2RE = \sqrt{\frac{\sum_{i=1}^n (y_i - y_i')^2}{\sum_{i=1}^n (y_i')^2}} = \frac{\|\mathbf{y} - \mathbf{y}'\|_2}{\|\mathbf{y}'\|_2}.$$

**What is (Adam + L-BFGS)$^2$?** One of the key factors contributing to the success of SAFE-NET in our experiments is the utilization of an effective hybrid optimizer, denoted as (Adam+L-BFGS)$^2$. This optimizer sequence is designed to leverage the strengths of both Adam and L-BFGS optimizers in a staged approach to achieve the best possible results for each problem.

The optimizer works as follows:

1. **Adam(1):** The first stage involves running Adam, referred to as Adam(1). This stage is used to approach a suitable region in the solution space, minimizing the risk of stalling in subsequent stages.

2. **L-BFGS(1):** After Adam(1), we employ L-BFGS (denoted as L-BFGS(1)), which utilizes the above settings.

3. **Adam(2):** When a stall in L-BFGS(1) is detected, Adam is run again (Adam(2)). This iteration of Adam is adaptive; the learning rate decreases by a predetermined factor every 2000 steps to fine-tune the approach towards the minimum.

4. **L-BFGS(2):** Another round of L-BFGS (L-BFGS(2)) is implemented at the end of the training process to rapidly converge to the solution.

## 3 RELATED WORK

Here we review common approaches for PINN training and feature engineering strategies proposed in the literature.

### 3.1 CHALLENGES IN LEARNING WITH PINNS

In order to learn effectively with a PINN model, it is crucial that the model be trained till it reachs a low loss — indeed, existing generalization bounds Mishra & Molinaro (2023) show the quality of the recovered solution increases as the training loss goes down. Unfortunately, training the vanilla PINN architecture is known to be extremely challenging Krishnapriyan et al. (2021); Rathore et al. (2024). On a fundamental level, this issue directly arises from of the presence of the differential operator in the loss. The differential operator can be highly ill-conditioned De Ryck et al. (2023), which leads to an ill-conditioned optimization landscape and slow convergence Rathore et al. (2024).

In order to address these challenges, a variety of modifications to the vanilla PINN have been proposed within the literature, many of which attempt to make the optimization problem easier to solve. Wang et al. (2021a; 2022a;b); Nabian et al. (2021); Wu et al. (2023a;b) propose loss reweighting/resampling to balance different components of the loss, Yao et al. (2023); Müller & Zeinhofer (2023) propose scale-invariant and natural gradient-based optimizers for PINN training, Jagtap et al. (2020a;b); Wang et al. (2023) propose adaptive activation functions which can accelerate convergence of the optimizer, and Liu et al. (2024) propose an approach to precondition the PINN loss itself. Other approaches include bespoke loss functions and regularizations E & Yu (2018); Lu et al. (2021); Kharazmi et al. (2021); Khodayi-Mehr & Zavlanos (2020); Yu et al. (2022) and new architectures Jagtap et al. (2020c); Jagtap & Karniadakis (2020); Li et al. (2020); Moseley et al. (2023). These strategies work with varying degrees of success, and no single strategy improves performance across all PDEs.

SAFE-NET differs from much of this prior work, in that it does not seek to directly change the optimization landscape. Instead, it seeks to better capture the inductive bias of the PDE, by augmenting the initial data representation with relevant features. However, we shall see that the better choice of representation yields can improve conditioning which leads to a better optimization landscape and fast training. Thus, SAFE-NET implicitly preconditions the problem with its feature engineering.

### 3.2 FEATURE ENGINEERING IN PINNS

We are not the first work to consider feature engineering PINNs. Several works have considered feature engineering in the context of spectral bias. Spectral bias refers to the inability of neural networks to learn high-frequency functions. Wang et al. (2021b) applies random Fourier features to enhance the network's ability to learn high-frequency functions more effectively, using their Fourier feature mapping. They use the feature mapping

$$\gamma(v) = [\cos(Bv), \sin(Bv)]$$

where $v$ denotes the input coordinates (space, time, etc.) and $B \in \mathbb{R}^{m \times d}$ is a matrix with elements from a Gaussian distribution $N(0, \sigma^2)$. By using both $\cos$ and $\sin$ components in their mapping, they try to have the network capture periodic, high-frequency behavior in the input data. This embedding is much less general then SAFE-NET, and they only explore the case when $B$ is normally distributed.

Another recent approach is the RBF-PINN (Burghardt et al.; Zeng et al., 2024). In the RBF-PINN method, the authors propose using Radial Basis Functions (RBF) instead of Fourier features. They use the RBF function:

$$\phi_{\text{RBF}}(x) = \exp\left(-\frac{|x - c|^2}{2\sigma^2}\right),$$

where $c$ is the center of the RBF, and $\sigma$ controls the width of the RBF. They argue that the RBF kernel can better handle problems with sharp changes or discontinuities, which Fourier features sometimes struggle with due to the Gibbs phenomenon. The restriction to RBF kernels is less general than SAFE-NET, and maybe more computationally intensive due to the need to perform kernel regression.

Moseley et al. (2023) proposed FBPINN, which models the solution as a sum of basis functions with neural network coefficients. Again this technique was developed to address spectral bias and the deficiencies of PINNs in capturing multiscale solutions. This approach can be viewed as a feature expansion of the solution, with the basis functions corresponding to features. Judicious choice of the basis functions can be viewed as feature engineering. This approach differs significantly from SAFE-NET, which engineers the input-features to the network to learn better solutions.

In total, most of the literature on feature engineering in PINNS has been focused on addressing spectral bias. Unlike SAFE-NET These works do not systematically investigate whether or not augmenting PINNS with physically relevant input features leads to significantly better performance.

## 4 THEORY

### 4.1 FOURIER FREQUENCIES AND FEATURE GENERATION

From Fourier theory, it is known that a function $f$ could be approximated using its dominant Fourier frequencies. In particular, $f(\mathbf{x}) \approx \sum_{\kappa \text{ dominant}} \left(A_\kappa \cos(2\pi\kappa \cdot \mathbf{x}) + B_\kappa \sin(2\pi\kappa \cdot \mathbf{x})\right)$. Additional details regarding these calculations are provided in Appendix B. There are different ways to utilize dominant frequencies in to generate suitable features for our network.

#### 4.1.1 INTEGER FOURIER FREQUENCIES

Let us first consider the two-dimensional case. Take

$$\{\cos(m\pi x - n\pi t), \cos(m\pi x + n\pi t), \sin(m\pi x - n\pi t), \sin(m\pi x + n\pi t)\}$$

to be a basis for $L^2([0,1]^2)$. Then, from the previous calculations, the function $f(x, t)$ can be approximated as:

$$\sum_{m,n} a_{mn} \cos(m\pi x - n\pi t) + b_{mn} \cos(m\pi x + n\pi t) + c_{mn} \sin(m\pi x - n\pi t) + d_{mn} \sin(m\pi x + n\pi t)$$

where $a_{mn}, b_{mn}, c_{mn}, d_{mn}$ are the Fourier coefficients that represent the amplitudes of the corresponding basis functions. Our experiments demonstrate that introducing a set of normalized Fourier features of the form

$$\{\cos(m\pi x - n\pi t), \cos(m\pi x + n\pi t), \sin(m\pi x - n\pi t), \sin(m\pi x + n\pi t)\}$$

into the network significantly improves the optimization of the loss function. Moreover, they establish that setting $m = n$ leads to sufficiently good results while keeping our optimization process fast. We further discuss and systematically compare feature selection strategies in Section 5.2.

### 4.1.2 "ORACLE"; DOMAIN KNOWLEDGE FEATURES

One potential approach involves calculating some dominant frequencies of $f$ and $\nabla f$ based on specific cases derived from the boundary and initial conditions of the PDE. This method is supported by the following mathematical lemma:

**Lemma.** The Fourier frequencies of $f$ and $\nabla f$ are identical, as given by:

$$\mathcal{F}\{\nabla f(\mathbf{x})\} = i\kappa \hat{f}(\kappa),$$

where $\mathcal{F}$ denotes the Fourier transform, $\nabla f$ represents the gradient of $f$, $\kappa$ is the frequency vector, and $\hat{f}$ is the Fourier transform of $f$.

Our experiments demonstrate that approximating the dominant frequencies of $f$ and $\nabla f$, especially in scenarios dictated by initial and boundary conditions, and employing these frequencies to generate Fourier features, significantly enhances the effectiveness of minimizing the loss function. An example illustrating this method is detailed in Section 5.2.

### 4.2 LOGARITHMIC AND POLYNOMIAL FEATURES

Our experiments show that, in addition to the previously discussed Fourier features, adding polynomial and logarithmic features to certain problems can improve loss function minimization, enhancing the error's order of magnitude. A key strength of logarithmic features is the identity:

$$\log(\omega x) = \log(\omega) + \log(x) \quad \text{for any } \omega \in \mathbb{R}$$

This property implies that adding $\log(\omega x)$ as a feature in the network is effectively the same as adding $\log(x)$ plus a constant term $\log(\omega)$. Since adding a constant has no impact on the network's performance, it suffices to include only $\log(x)$ and $\log(t)$ as features. In certain problems, these logarithmic features could make a substantial difference.

We also found that incorporating polynomial features could be beneficial in specific scenarios. For our experiments, we added $x^2$, $xt$, and $t^2$, as polynomials of higher degrees do not seem to contribute to further noticeable improvements in loss function minimization in our experiments.

## 5 EXPERIMENTS

In this section, we validate our feature engineering approach for PINNs across a number of PDEs. Our experiments are structured into three main sections.

### 5.1 A COMPARISON BETWEEN NETWORK STRUCTURES AND OPTIMIZERS

We first compare different network architectures and optimizers with and without added features. Figure 1 illustrates the performances of different optimizers on three network structures for each PDE. For all these problems, we employ the basic feature set $S = \{\sin(\pi x), \sin(\pi t), \cos(\pi x), \cos(\pi t), \sin(x - t), \sin(x + t), \cos(x - t), \cos(x + t)\}$, where each feature is normalized before being fed through the single layer network. Each optimizer runs for 20,000 epochs.

We demonstrate that all our tested optimizers—Adam, L-BFGS, Adam+L-BFGS, and (Adam+L-BFGS)$^2$—perform optimally in SAFE-NET. Particularly, (Adam+L-BFGS)$^2$ shows a remarkable improvement over the other optimizers, enhancing the error's order of magnitude by one or two orders in each problem. Further experiments indicate that increasing the number of layers while maintaining the feature set $S$ results in a negligible difference; these results along with the final loss table of Figure 1 (Table 4) are detailed in Appendix C.

Table 1 summarizes the L2 relative error (L2RE) of each optimizer for each network structure. Notably, (Adam+L-BFGS)$^2$ on SAFE-NET achieves $21\times$ smaller L2RE than the best PINN L2RE

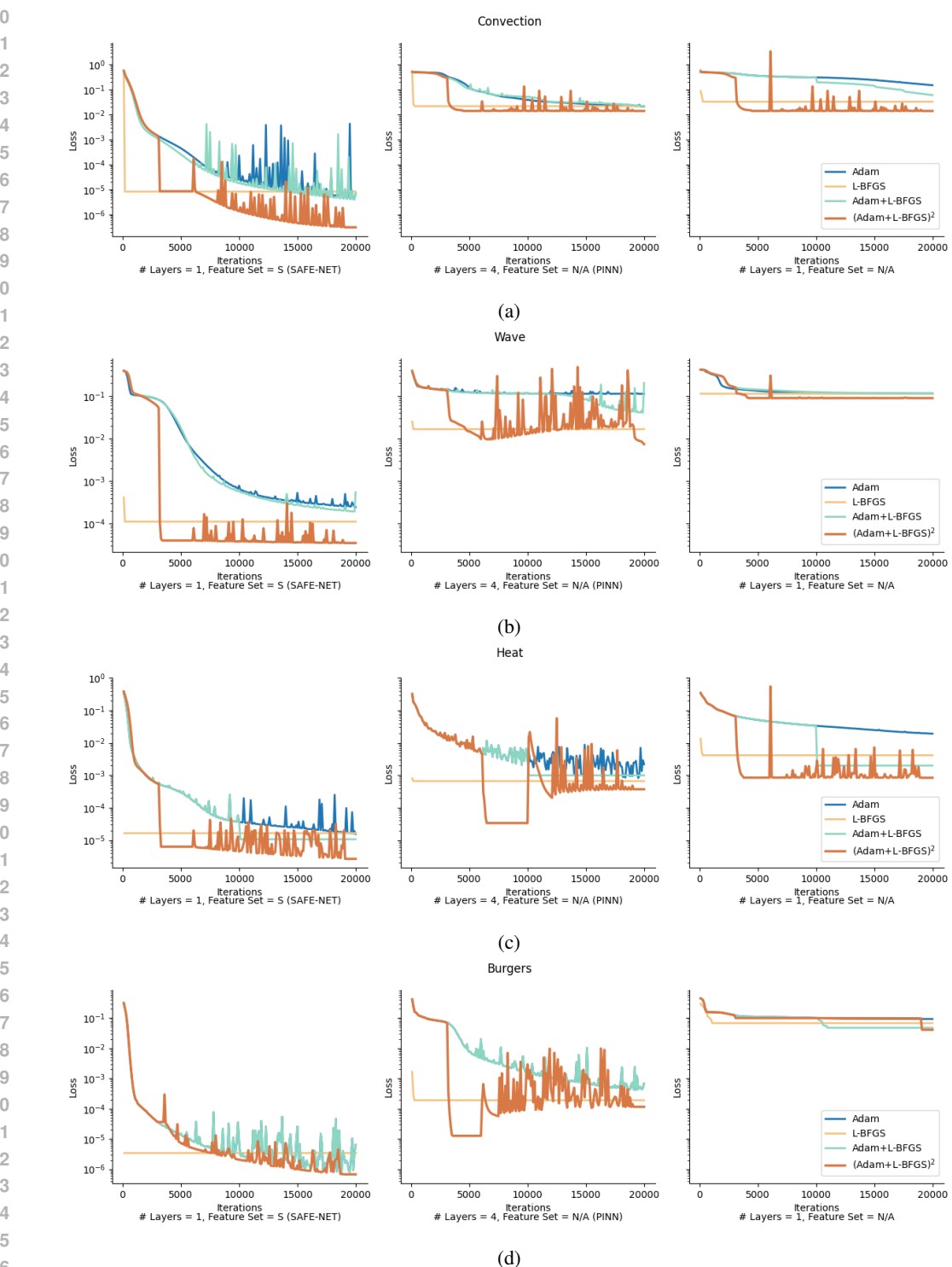

Figure 1: Performance of Adam, L-BFGS, Adam + L-BFGS, and (Adam + L-BFGS)$^2$ on each problem. The structure set-up is mentioned below each figure.

result on the Wave problem, $323\times$ smaller L2RE than the best PINN result on the Convection problem, and more than $6\times$ smaller L2RE than the best PINN result on the other problems for only 20,000 epochs.

Table 1: L2RE for different network structures and optimizers for the wave (W), convection (C), heat (H), and Burger's (B) PDEs after 20k epochs

| | Structure | | Optimizer | | | |
|---|---|---|---|---|---|---|
| | # Layers | Feature Set | Adam | L-BFGS | Adam+L-BFGS | $(\text{Adam+L-BFGS})^2$ |
| W | 4 | N/A | 9.03e-01 | 2.84e-01 | 5.55e-01 | 2.27e-01 |
| | 1 | N/A | 8.68e-01 | 8.79e-01 | 8.99e-01 | 7.85e-01 |
| | 1 | S | 3.26e-02 | 2.74e-02 | 3.40e-02 | **1.05e-02** |
| C | 4 | N/A | 4.00e-01 | 4.02e-01 | 4.05e-01 | 3.13e-01 |
| | 1 | N/A | 1.04e+00 | 4.89e-01 | 8.26e-01 | 2.97e-01 |
| | 1 | S | 1.92e-02 | 1.03e-02 | 8.87e-03 | **9.67e-04** |
| H | 4 | N/A | 7.83e-02 | 7.04e-02 | 7.33e-02 | 1.58e-02 |
| | 1 | N/A | 3.78e-01 | 1.76e-01 | 1.57e-01 | 8.57e-02 |
| | 1 | S | 1.72e-02 | 1.42e-02 | 2.15e-02 | **1.17e-03** |
| B | 4 | N/A | 7.09e-02 | 5.40e-02 | 5.66e-02 | 4.91e-03 |
| | 1 | N/A | 8.33e-01 | 7.44e-01 | 6.20e-01 | 5.89e-01 |
| | 1 | S | 2.62e-03 | 5.09e-03 | 2.46e-03 | **9.18e-04** |

## 5.2 A COMPARISON BETWEEN FEATURE ENGINEERING METHODS

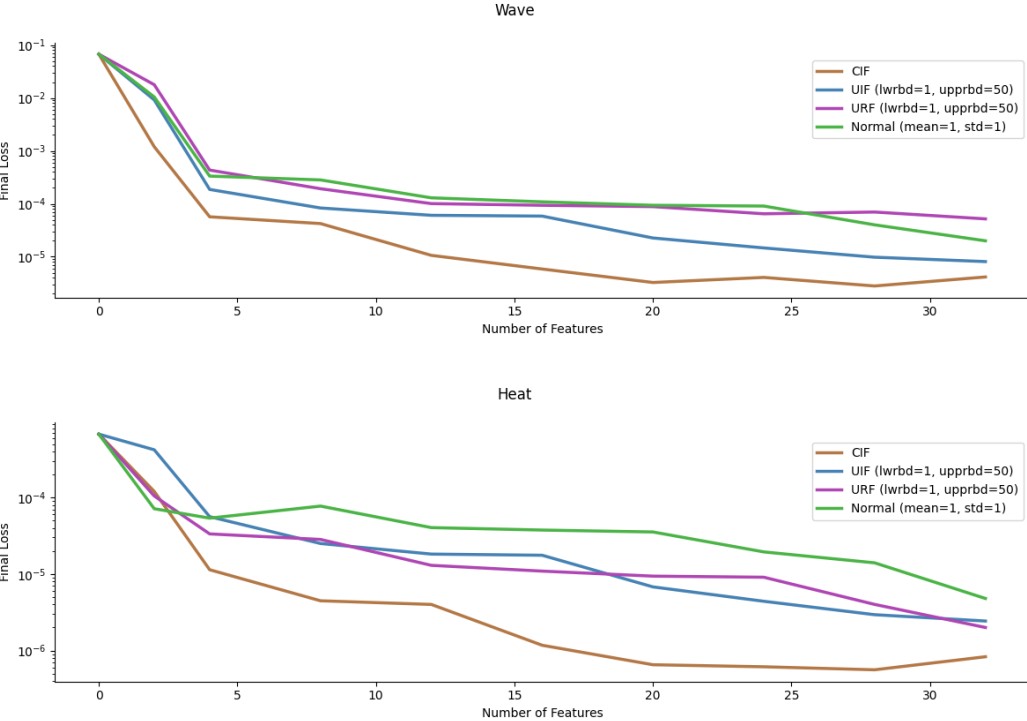

Figure 2: Performance of each feature generation method on the wave and heat PDEs

In this section, we test our hypothesis by exploring how different feature engineering techniques affect our loss function optimization. We use SAFE-NET with S as well as additional features. We employ the following methods to choose the additional features:

1. **Consecutive Integer Features (CIF):** We define an integer upper bound $M$ and consider all integers $n \in \{1, 2, 3, \ldots, M\}$. Using these integers, we construct the feature set

$$S_M = S \cup \bigcup_{k=2}^{M} \{\sin k(x - t), \sin k(x + t), \cos k(x - t), \cos k(x + t)\}$$

   as our new expanded feature set. For example, $S_1 = S, S_2 = S \cup \{\sin 2(x - t), \sin 2(x + t), \cos 2(x - t), \cos 2(x + t)\}$ and so forth, up to $S_M$.

2. **Uniformly-Chosen Integer Features (UIF):** We fix an integer lower bound $M_{\min}$ and an integer upper bound $M_{\max}$, and $n$ as the number of features. We then choose $n$ integers using the uniform distribution $U(M_{\min}, M_{\max})$. We form the set

$$S_{M_{\min}, M_{\max}} = S \cup \bigcup \{\sin k(x - t), \sin k(x + t), \cos k(x - t), \cos k(x + t)\},$$

   where $k \sim U(M_{\min}, M_{\max})$, as our new feature set.

3. **Uniformly-Chosen Real Features (URF):** Similar to the above, but $k$ is taken to be a uniformly chosen real number from $U(M_{\min}, M_{\max})$, where $M_{\min}$ and $M_{\max}$ are the real lower and upper bound respectively.

4. **Other distributions:** We also utilize a normal distribution to pick our frequencies, which proves to be less effective as it centers around a specific number and does not provide the diversity. This is the complete opposite of our intuitive Fourier basis approach.

5. **Domain Knowledge Features (DKF):** In many instances, the initial and boundary conditions of a Partial Differential Equation (PDE) carry simple yet valuable information that can be directly used as features. We detail the training results utilizing DKF in subsequent plots and tables, referred to as "oracle". An example illustrating the practical application of DKF will be presented later in this section to better demonstrate the use of DKF.

Figure 2 illustrates a comparison of approaches 1 through 4 from an approximation theory perspective on the wave and heat problems, where a decrease in the error order of magnitude is spotted as the number of features increases. Each plot within the figure displays the final loss vs. the number of features generated using each approach on each PDE.

**Example of DKF.** In addition to studying these feature selection methods, we could also incorporate DKF relevant to each specific PDE as previously discussed. To demonstrate this method, let us use the wave PDE as an example. From the description of the wave problem in the appendix, we have the initial condition:

$$u(x, 0) = \sin(\pi x) + \frac{1}{2}\sin(\beta \pi x), \quad x \in [0, 1],$$

where $\beta = 5$ in our experiments. Based on this information, we select our DKF to be the set

$$S_{DKF} = \{\sin(\pi x), \sin(\pi t), \cos(\pi x), \cos(\pi t), \sin(5\pi x), \sin(5\pi t), \cos(5\pi x), \cos(5\pi t)\}.$$

We train our model using the usual single-layer network, feeding the eight features of set $S_{DKF}$ through. The results are provided in Figure G, where the DKF "oracle" is displayed by a dashed line to be easily distinguishable. Figure G shows that the DKF of our choice was able to perform even better than CIF, reaching an error order of $10^{-7}$ for the wave problem; Figure 3.

However, in many problems, the initial and boundary conditions may not provide useful information for feature selection. Contrary to this approach, a significant advantage of the feature engineering methods we presented is their general applicability as well as simplicity.

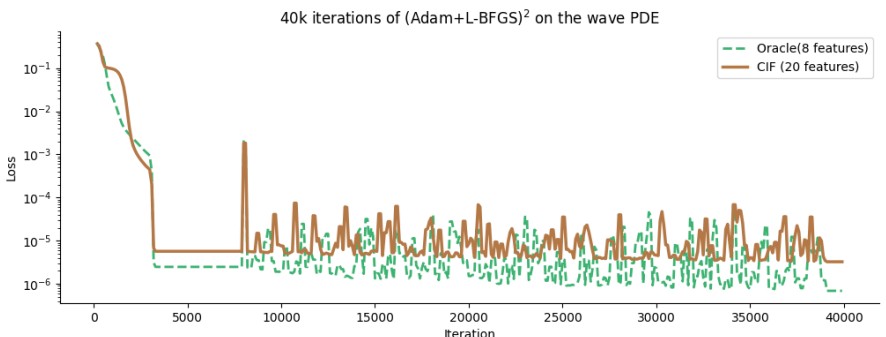

Figure 3: Comparison of CIF with 20 features and DKF for the wave problem

**Logarithmic and polynomial features could be helpful.** Now, we present the results of our experiments on the reaction PDE. These results are presented separately because, with this problem, we demonstrate the effectiveness of a slightly modified basic feature set $S'$, as discussed in Section 4.2. We define

$$S' = \{x^2, xt, t^2, \log x, \log t\} \cup S$$

and use $S'$ as our basic feature set to achieve the best performance. We generate new features based on $S'$ in the same manner as with $S$ using CIF, UIF, URF, and other distributions. For instance, using CIF for a given $M$, we would have

$$S'_M = S' \cup \bigcup_{k=2}^{M} \{\sin k(x - t), \sin k(x + t), \cos k(x - t), \cos k(x + t)\}$$

and so forth. We experimented with both $S$ and $S'$ as our basic feature sets, and the results are provided in Figure 4.

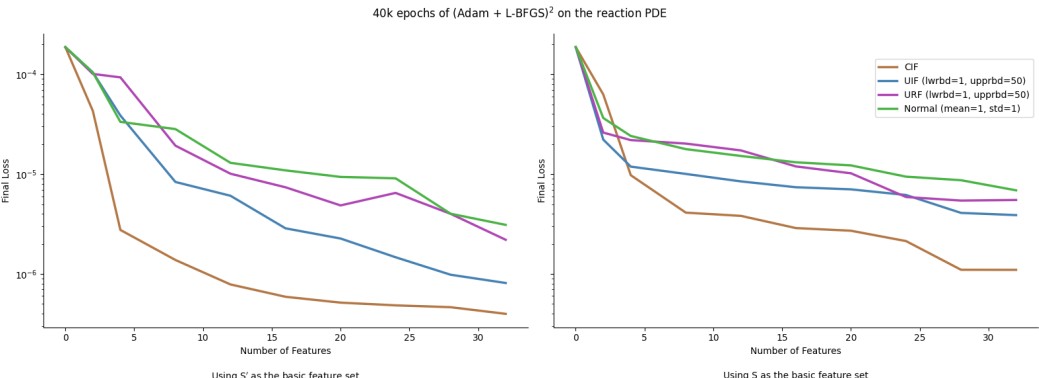

Figure 4: Final loss vs. number of features for the reaction problem. The figure on the left uses $S'$ as the basic feature set while the figure on the right uses $S$. Evidently, having polynomial and logarithmic features helps in improving the optimization.

Table 2 summarizes the best performance of each feature generation method across all tests conducted using $(\text{Adam} + \text{L-BFGS})^2$. As evident from the table, using CIF, we achieve significantly lower losses. For the Wave problem, for instance, we reach an error order of $10^{-6}$ with as few as 16 generated features—or even fewer if we select an effective subset of these 16 features. This high performance is achieved very quickly, requiring only a few iterations using Adam+L-BFGS, as detailed in Figure 3. Comparing these results with those presented in Figure 1 and Table 1, we observe a significant improvement, underscoring the effectiveness of our feature engineering strategy.

Table 2: Final loss and L2RE using SAFE-NET with (Adam + L-BFGS)$^2$ for the wave (W), convection (C), heat (H), and Burger's (B) PDEs after 40k epochs for different feature engineering strategies

| | | Normal | URF | UIF | CIF | DKF |
|---|---|---|---|---|---|---|
| Wave | Loss | 6.43e-05 | 8.98e-05 | 8.19e-06 | 4.14e-6 | **6.32e-07** |
| | L2RE | 2.17e-02 | 2.94e-02 | 9.73e-03 | 5.49e-03 | **2.15e-03** |
| Convection | Loss | 1.80e-06 | 6.01e-07 | 3.14e-07 | **2.11e-07** | - |
| | L2RE | 3.62e-03 | 1.03e-03 | 9.94e-04 | **8.69e-04** | - |
| Heat | Loss | 9.80e-06 | 6.01e-06 | 5.14e-06 | **4.94e-07** | - |
| | L2RE | 1.03e-02 | 5.15e-03 | 7.03e-03 | **1.04e-03** | - |
| Burgers | Loss | 8.47e-06 | 7.21e-06 | 7.49e-07 | **4.24e-07** | - |
| | L2RE | 9.60e-03 | 7.86e-03 | 1.68e-03 | **9.07e-04** | - |
| Reaction | Loss | 7.44e-06 | 5.23e-06 | 9.79e-07 | **4.18e-07** | - |
| | L2RE | 1.46e-02 | 1.01e-02 | 8.87e-03 | **7.32e-03** | - |

## 5.3 A COMPARISON BETWEEN INITIALIZATION AND PROBLEM CONDITIONING

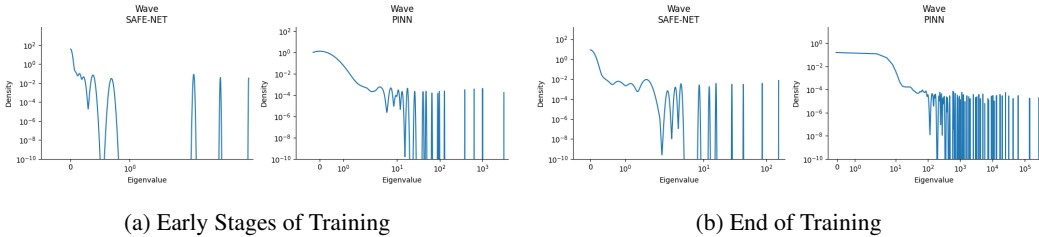

(a) Early Stages of Training                     (b) End of Training

Figure 5: Spectral density plots for the wave PDE at different stages of training

We conduct a comparison between the conditioning of each problem both at the early stages (after 5000 epochs of Adam) and after training using SAFE-NET and PINN. We utilize (Adam+L-BFGS)$^2$ for 40,000 iterations, using the optimizer settings of Section 5.1 as illustrated in Figure 5 for the wave PDE. Similar spectral density comparison plots for all other PDEs are provided in Appendix C. We observe considerable improvements in the conditioning of the problems even at the initial phase, which suggests that SAFE-NET does more effective initialization as well. Additionally, the spectral density plots at the end of the training period indicate dramatic improvements in the conditioning of each problem using SAFE-NET. In particular, the top eigenvalues for the wave and convection problems are reduced by a factor of $10^3$, the top eigenvalues for the heat and Burgers problems are reduced by a factor of $10^2$. Overall, we observe a significant reduction in the number and density of large eigenvalues in each problem as well.

## 6 CONCLUSION

This study presented SAFE-NET, a simplified single-layer architecture enhanced by feature engineering, addressing the computational and performance challenges commonly associated with Physics-Informed Neural Networks (PINNs). By employing the (Adam + L-BFGS)$^2$ optimizer, SAFE-NET efficiently reduces the prediction error by orders of magnitude using far fewer parameters compared to traditional multi-layer PINN architectures. Our results confirm that SAFE-NET not only expedites the training process but also improves problem conditioning and generalization across various PDE challenges. These findings advocate for a reevaluation of network complexity in PINNs, highlighting the potential for achieving significant advancements in PDE solutions through more efficient and focused network designs.

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

APPENDIX

# A  ADDITIONAL DETAILS ON PROBLEM SETUP

In this section of the appendix, we present the differential equations we study in our experiments.

## A.1  PHYSICS-INFORMED NEURAL NETWORKS

Physics-Informed Neural Networks (PINNs) are a class of neural networks that incorporate physical laws described by Partial Differential Equations (PDEs) into the training process. PINNs solve forward and inverse problems involving PDEs by embedding the physics constraints into the loss function. They aim to solve PDE systems of the form:

$$D[u(x), x] = 0, \quad x \in \Omega$$
$$B[u(x), x] = 0, \quad x \in \partial\Omega$$
$$I[u(x), x] = 0, \quad x \in \Omega$$

Where $D$ represents the differential operator defining the PDE, $B$ represents the boundary conditions. $I$ represents the initial conditions, important for time-dependent problems, and $\Omega \subseteq \mathbb{R}^n$ is the domain of the PDE.

**Loss Function in PINNs.** PINNs minimize a non-linear least-squares loss consisting of three terms:

$$L(w) = \frac{1}{2n_{\text{res}}} \sum_{i=1}^{n_{\text{res}}} \left(D[u(x_i^r; w), x_i^r]\right)^2 + \frac{1}{2n_{\text{bc}}} \sum_{j=1}^{n_{\text{bc}}} \left(B[u(x_j^b; w), x_j^b]\right)^2 + \frac{1}{2n_{\text{ic}}} \sum_{k=1}^{n_{\text{ic}}} \left(I[u(x_k^i; w), x_k^i]\right)^2$$

here the first term $(D)$ represents the PDE residual loss, the second term $(B)$ represents the boundary condition loss, and the third term $(I)$ ensures the initial condition loss for time-dependent problems.

## A.2  WAVE

The wave equation, a type of hyperbolic partial differential equation (PDE), is commonly encountered in the study of phenomena such as acoustics, electromagnetism, and fluid dynamics. Our focus is on the following wave equation:

$$\frac{\partial^2 u}{\partial t^2} - 4\frac{\partial^2 u}{\partial x^2} = 0, \quad x \in (0,1), \ t \in (0,1),$$

with the initial conditions:

$$u(x,0) = \sin(\pi x) + \frac{1}{2}\sin(5\pi x), \quad x \in [0,1],$$

$$\frac{\partial u(x,0)}{\partial t} = 0, \quad x \in [0,1],$$

and boundary conditions:

$$u(0,t) = u(1,t) = 0, \quad t \in [0,1].$$

The analytical solution for this PDE, setting $\beta = 5$, is given by $u(x,t) = \sin(\pi x)\cos(2\pi t) + \frac{1}{2}\sin(5\pi x)\cos(10\pi t)$.

## A.3  CONVECTION

The convection equation, another hyperbolic PDE, models processes such as fluid flow, heat transfer, and biological dynamics. We examine this equation:

$$\frac{\partial u}{\partial t} + \beta\frac{\partial u}{\partial x} = 0, \quad x \in (0, 2\pi), \ t \in (0,1),$$

with the initial condition:

$$u(x,0) = \sin(x), \quad xin[0, 2\pi],$$

and the cyclic boundary condition:

$$u(0,t) = u(2\pi, t), \quad t \in [0,1].$$

The exact solution to this equation with $\beta = 40$ is $u(x,t) = \sin(x - 40t)$.

### A.4 HEAT

The heat equation is fundamental in the mathematical modeling of thermal diffusion processes. It is widely applied in fields such as thermodynamics, material science, and environmental engineering to analyze heat distribution over time within solid objects. This equation is also crucial in under-standing temperature variations in earth sciences, predicting weather patterns in meteorology, and simulating cooling processes in manufacturing industries. We study this parabolic PDE, expressed as:

$$\frac{\partial u}{\partial t} - 4\frac{\partial^2 u}{\partial x^2} = 0, \quad x \in [0, 2], \ t \in [0, 0.2],$$

with the initial profile:

$$u(x, 0) = x^2(2 - x), \quad x \in [0, 2],$$

and fixed boundary conditions:

$$u(0, t) = u(2, t) = 0, \quad t \in [0, 0.2].$$

The experiments use $\kappa = 2$.

### A.5 BURGERS

The Burgers equation, a fundamental partial differential equation (PDE) in fluid mechanics, is used to model various nonlinear phenomena including shock waves and traffic flow. We examine the following form of the Burgers' equation:

$$\frac{\partial u}{\partial t} + u\frac{\partial u}{\partial x} = \nu\frac{\partial^2 u}{\partial x^2}, \quad x \in [-1, 1], \ t \in [0, 1],$$

where $\nu = \frac{0.01}{\pi}$ represents the viscosity, crucial for modeling the diffusion effects.

The boundary conditions are periodic:

$$u(-1, t) = u(1, t), \quad t \in [0, 1],$$

and the initial condition is given by:

$$u(x, 0) = -\sin(\pi x), \quad x \in [-1, 1].$$

The analytical solution to this PDE, which can be derived under certain conditions, represents the evolution of the wave profile influenced by both convection and diffusion. This equation helps illustrate the balance between nonlinear advection and viscosity, essential for understanding the dynamics of the modeled system.

### A.6 REACTION

The reaction equation, a nonlinear ordinary differential equation (ODE), is useful for modeling chemical kinetics. We analyze it under the conditions:

$$\frac{\partial u}{\partial t} - 5u(1 - u) = 0, \quad x \in (0, 2\pi), \ t \in (0, 1),$$

$$u(x, 0) = \exp\left(-\frac{(x - \pi)^2}{2(\pi/4)^2}\right), \quad x \in [0, 2\pi],$$

$$u(0, t) = u(2\pi, t), \quad t \in [0, 1].$$

The solution formula for this ODE with $\rho = 5$ is expressed as $u(x, t) = \frac{h(x)e^{5t}}{h(x)e^{5t}+1-h(x)}$, where $h(x) = \exp\left(-\frac{(x-\pi)^2}{2(\pi/4)^2}\right)$.

# B  ADDITIONAL THEORETICAL REMARKS

## B.1  PRELIMINARIES; DOMINANT FREQUENCIES

From Fourier theory, it is known that the Fourier expansion of a periodic single variable function $f(t)$ can be expressed as follows:

$$f(t) = \sum_{n=0}^{\infty} (a_n \cos(n\omega_0 t) + b_n \sin(n\omega_0 t))$$

where $\omega_0$ is the fundamental angular frequency, $T$ is the period of the function, and $a_n$ and $b_n$ are the Fourier coefficients. Similarly, to express a two-variable periodic function $f(x, t)$ using Fourier series, we represent it as a sum of sinusoidal functions in both the spatial variable $x$ and the temporal variable $t$ as

$$f(x, t) = \sum_{m=0}^{\infty} \sum_{n=0}^{\infty} \Big[ a_{mn} \cos(m\kappa_0 x - n\omega_0 t) + b_{mn} \cos(m\kappa_0 x + n\omega_0 t)$$

$$+ c_{mn} \sin(m\kappa_0 x - n\omega_0 t) + d_{mn} \sin(m\kappa_0 x + n\omega_0 t)) \Big].$$

Here, $a_{mn}, b_{mn}, c_{mn}, d_{mn}$ are the Fourier coefficients, which can be computed based on the function $f(x, t)$. Moreover, $\kappa_0$ represents the spatial frequency and $\omega_0$ represents the temporal frequency. However, in the general case, when the function $f(x, t)$ is not periodic in $x$ or $t$, we would instead use the continuous Fourier transform,

$$f(x, t) = \int_{-\infty}^{\infty} \int_{-\infty}^{\infty} \Big[ A(k, \omega) \cos(m\kappa x - n\omega t) + B(k, \omega) \cos(m\kappa x + n\omega t)$$

$$+ C(k, \omega) \sin(m\kappa x - n\omega t) + D(k, \omega) \sin(m\kappa x - n\omega t) \Big] dk \, d\omega.$$

In general, the function $f(\mathbf{x})$ can be reconstructed from its Fourier transform using the inverse Fourier transform given by

$$f(\mathbf{x}) = \int_{-\infty}^{\infty} \hat{f}(\kappa) e^{2\pi i \kappa \cdot \mathbf{x}} \, d\kappa.$$

Using this formulation, one can attempt to approximate $f(\mathbf{x})$ using only the "dominant frequencies", which are frequencies $\kappa$ where $|\hat{f}(\kappa)|$ is large. By summing only over these dominant frequencies, we get $f(\mathbf{x}) \approx \sum_{\kappa \text{ dominant}} \hat{f}(\kappa) e^{2\pi i \kappa \cdot \mathbf{x}}$. Since $\hat{f}(\kappa)$ can be complex, it can be expressed as

$$f(\mathbf{x}) \approx \sum_{\kappa \text{ dominant}} (A_\kappa \cos(2\pi \kappa \cdot \mathbf{x}) + B_\kappa \sin(2\pi \kappa \cdot \mathbf{x})).$$

Here, $A_\kappa$ and $B_\kappa$ are real coefficients derived from $\hat{f}(\kappa)$. Clearly, any information on dominant frequencies $\kappa$ and coefficients $A_\kappa$ and $B_\kappa$ would help us better approximate $f$.

**Remark on the number of frequencies.** Suppose $\mathbf{x} = (x_1, x_2, \ldots, x_n)$ is an $n$-dimensional vector, and we aim to choose features to approximate a function $f(\mathbf{x})$. Each dimension can be associated with a sine or cosine function at a particular frequency $\omega_i$. For a fixed set of frequencies, this setup yields $2^n$ combinations of sine and cosine functions. When allowing the frequencies to vary within a set such as $\{1, 2, \ldots, M\}$, the number of possible frequency combinations for each function would be $(2M)^n$, illustrating an exponential growth in the number of potential features. This exponential increase could become computationally challenging for problems with high dimensions. However, this approach offers significant advantages for lower-dimensional problems, where the comprehensive coverage of frequency space can greatly enhance the approximation quality of $f(\mathbf{x})$.

# C  ADDITIONAL EXPERIMENTAL RESULTS

## C.1  IMPACT OF INCREASING THE NUMBER OF LAYERS WITH FEATURE SET $S$

As discussed in Section 5.1 of Experiments, Table 4 summarizes the best performance of each optimizer on different network structures—single-layer without features, PINN, and SAFE-NET—over

20,000 epochs. In this section, we provide further in Table 3 results concerning additional network structures, demonstrating that when employing the feature set $S$, increasing the number of neural network layers does not significantly enhance the final result. Consequently, using fewer layers, which implies fewer parameters, renders SAFE-NET substantially faster and more efficient.

Table 3: Final loss for multi-layer neural networks paired with feature set S and various optimizers, applied to wave (W), convection (C), heat (H), and Burger's (B) PDEs after 20,000 epochs, showing negligible difference

| | Structure | | Optimizer | | | |
|---|---|---|---|---|---|---|
| | # Layers | Feature Set | Adam | L-BFGS | Adam+L-BFGS | $(\text{Adam+L-BFGS})^2$ |
| W | 4 | S | 1.14e-04 | 1.11e-04 | 1.01e-04 | 2.76e-05 |
| | 3 | S | 1.43e-04 | 1.23e-04 | 1.18e-04 | 4.92e-05 |
| | 2 | S | 1.32e-04 | 1.27e-04 | 1.23e-04 | 1.96e-05 |
| C | 4 | S | 3.99e-06 | 7.26e-06 | 2.12e-06 | 4.39e-07 |
| | 3 | S | 9.47e-06 | 1.26e-05 | 5.26e-6 | 1.01e-06 |
| | 2 | S | 3.71e-06 | 6.43e-06 | 4.44e-06 | 2.46e-7 |
| H | 4 | S | 1.74e-04 | 9.39e-05 | 8.43e-05 | 1.98e-06 |
| | 3 | S | 6.98e-05 | 2.79e-05 | 4.66e-05 | 5.11e-06 |
| | 2 | S | 3.31e-05 | 4.39e-05 | 1.02e-05 | 1.45e-06 |
| B | 4 | S | 9.78e-07 | 8.98e-07 | 7.17e-07 | 6.55e-07 |
| | 3 | S | 5.98e-06 | 6.35e-06 | 9.87e-07 | 9.17e-07 |
| | 2 | S | 4.12e-07 | 1.19e-06 | 5.32e-07 | 4.95e-07 |

## C.2 FINAL LOSS TABLE FOR SECTION 5.1

Table 4 summarizes the best performance of each optimizer on each network structure over 20,000 epochs. Note that $S$ represents the basic feature set defined at the beginning of this section. Once again, $(\text{Adam+L-BFGS})^2$ demonstrates superior performance compared to using Adam+L-BFGS or either Adam or L-BFGS alone.

Table 4: Final loss for different network structures and optimizers for the wave (W), convection (C), heat (H), and Burger's (B) PDEs after 20k epochs

| | Structure | | Optimizer | | | |
|---|---|---|---|---|---|---|
| | # Layers | Feature Set | Adam | L-BFGS | Adam+L-BFGS | $(\text{Adam+L-BFGS})^2$ |
| W | 4 | N/A | 1.12e-01 | 1.15e-02 | 4.07e-02 | 7.42e-03 |
| | 1 | N/A | 1.23e-01 | 1.12e-01 | 1.18e-01 | 8.92e-02 |
| | 1 | S | 1.93e-04 | 1.12e-04 | 1.93e-04 | **3.53e-05** |
| C | 4 | N/A | 2.19e-02 | 2.16e-02 | 2.22e-02 | 1.39e-02 |
| | 1 | N/A | 1.49e-01 | 3.26e-02 | 9.53e-02 | 1.25e-02 |
| | 1 | S | 4.82e-06 | 8.11e-06 | 4.97e-06 | **3.06e-7** |
| H | 4 | N/A | 8.23e-04 | 6.67e-04 | 7.26e-04 | 3.37e-05 |
| | 1 | N/A | 1.92e-02 | 4.15e-03 | 3.35e-03 | 9.89e-04 |
| | 1 | S | 3.95e-05 | 2.69e-05 | 1.78e-05 | **2.45e-06** |
| B | 4 | N/A | 6.82e-04 | 3.91e-04 | 4.33e-04 | 2.15e-05 |
| | 1 | N/A | 9.35e-02 | 7.43e-02 | 5.13e-02 | 4.67e-02 |
| | 1 | S | 9.21e-07 | 3.47e-06 | 8.12e-07 | **5.65e-07** |

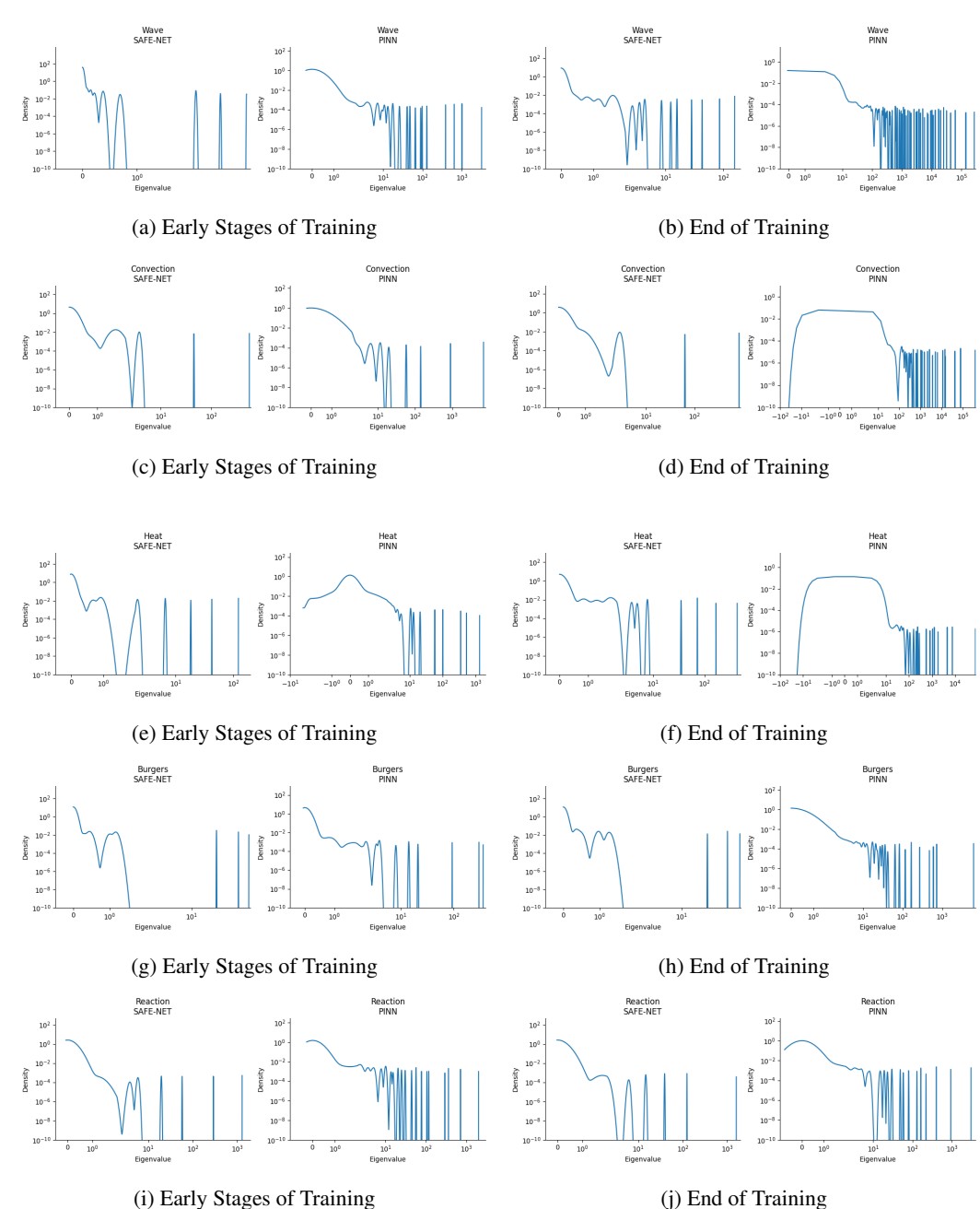

Figure 6: Spectral density plots for the wave, convection, heat, Burgers, and reaction problems at different stages of training.

## C.3 SPECTRAL DENSITY PLOT COMPARISON FOR SECTION 5.3

Figures 6a, 6c, 6e, 6g, and 6i display the spectral density plots at the early stages of training for the wave, convection, heat, Burgers, and reaction problems respectively, indicating considerable improvements in the conditioning of the problems even at this initial phase, which suggests that SAFE-NET possesses a more efficient and better initialization as well. Additionally, Figures 6b, 6d, 6f, 6h, and 6j present the spectral density plots at the end of the training period for each of these problems. Again, we observe dramatic improvements in the conditioning of each problem using SAFE-NET. In particular, the top eigenvalues for the wave and convection problems are reduced by a factor of $10^3$, the top eigenvalues for the heat and Burgers problems are reduced by a factor of $10^2$.

Overall, we observe a significant reduction in the number and density of large eigenvalues in each problem as well.

