# OpenReview forum: "Enhancing Physics-Informed Neural Networks Through Feature Engineering"
_ICLR.cc/2025/Conference — ICLR 2025 Conference Withdrawn Submission_

### Official Review · Reviewer_WQr6 · 2024-10-31

**Soundness:** 2
**Presentation:** 2
**Contribution:** 2
**Rating:** 3
**Confidence:** 5

**Summary:**

This paper proposes SAFE-NET (Single-layered Architecture with Feature Engineering), a simplified approach to Physics-Informed Neural Networks (PINNs) for solving partial differential equations. The key insight is that carefully engineered input features combined with a simple network architecture can outperform conventional deep PINNs that rely on multiple layers to learn features.

The authors present a simplified PINN architecture that may only use a single layer combined with engineered features. Despite its light parametrization, SAFE-NET achieves good results in a small number of training epochs compared to traditional multi-layer PINNs. To optimize this architecture, the authors introduce a new optimization strategy called (Adam + L-BFGS)² that combines Adam and L-BFGS optimizers in a specific sequence: Adam(1) → L-BFGS(1) → Adam(2) with adaptive learning rate → L-BFGS(2).

For feature engineering, the paper presents several approaches including Consecutive Integer Features (CIF), Uniformly-Chosen Integer Features (UIF), Uniformly-Chosen Real Features (URF), and Domain Knowledge Features (DKF) derived from PDE boundary conditions. The authors also explore additional polynomial and logarithmic features for certain problems.

The work includes empirical validation across multiple PDE types including wave, convection, heat, Burgers, and reaction equations. Through these experiments, the authors aim to demonstrate SAFE-NET's performance in terms of prediction errors, convergence speed, problem conditioning throughout training, and parameter efficiency. Their results hope to challenge the belief that deep architectures are necessary for PINNs to learn effective features, showing instead that a simpler architecture with well-chosen features can be more effective.

**Strengths:**

The paper's primary strength lies in its comprehensive demonstration that engineered features can significantly enhance PINN performance. While this concept isn't novel and has been explored in previous literature, the authors contribute by providing a systematic evaluation of different feature engineering approaches and their combinations. Their empirical investigation includes comparative analysis of Consecutive Integer Features (CIF), Uniformly-Chosen Integer Features (UIF), Uniformly-Chosen Real Features (URF), and Domain Knowledge Features (DKF), offering empirical insights into the relative effectiveness of each approach.


The work presents empirical evidence supporting the efficacy of engineered features in PINNs. Through experiments across multiple PDE types (wave, convection, heat, Burgers, and reaction equations), the authors demonstrate that their simplified architecture with engineered features can achieve comparable or better results than traditional deep MLP networks while using significantly fewer parameters.


Another contribution is the introduction of the (Adam + L-BFGS)² optimization protocol. This structured sequence of optimization steps (Adam(1) → L-BFGS(1) → Adam(2) with adaptive learning rate → L-BFGS(2)) shows consistent performance improvements across all tested PDEs. The authors support this finding with empirical results demonstrating faster convergence and better final solutions compared to conventional optimization approaches in PINNs.

**Weaknesses:**

While the key idea of using feature engineering for improving the performance of PINNs has merit, I have several major concerns about the current presentation and experimentation presented in the manuscript:


1. The experimental comparisons rely heavily on vanilla MLP networks, which are known to be a very weak baseline for PINNs. The absence of comparisons against state-of-the-art methods like PirateNets [1] makes it difficult to assess the true significance of SAFE-NET's contributions. A more compelling evaluation would combine SAFE-NET's feature embeddings and optimization protocol with architectures like PirateNets to determine if they can achieve state-of-the-art results. This would provide stronger evidence for the generalizability of the proposed approach.


2. The paper's evaluation is limited to PDEs with smooth solutions, which represent relatively simple benchmarks in the current PINN literature. While these problems do highlight basic PINN training challenges, recent work has introduced substantially more challenging benchmarks where competitive results have been achieved by other methods [1,2]. The authors should extend their evaluation to these more demanding scenarios, such as problems with discontinuities, sharp gradients, or multi-scale features, to properly demonstrate SAFE-NET's capabilities against the current state-of-the-art. More concretely, I suggest that you consider evaluating your method against SOTA PINNs methods likePirateNets on the following benchmarks: KS, Gray-Scott and Ginzburg-Landau (see [1,2]).


3. The feature map selection process appears to lack a rigorous foundation. The authors switch between different feature types (e.g., Fourier embeddings for wave equations, logarithmic and polynomial features for reaction PDEs) without providing clear mathematical justification for these choices. A significant enhancement would be developing a systematic methodology for selecting feature maps based on PDE properties. For instance, with linear PDEs, one would expect optimal feature maps to correspond to PDE operator eigenfunctions. Without such theoretical grounding, the feature selection process remains largely heuristic and it is hard for me to see it achieve practical impact.


4. The (Adam + L-BFGS)² optimization protocol, while showing some empirical improvements, lacks theoretical justification. The paper would be strengthened by providing mathematical intuition or analysis explaining why this particular sequence of optimizers is effective. The current results, while promising, are limited to simple PDEs and weak baselines (see comments above). This is particularly concerning given that some state-of-the-art PINN results have been achieved using standard Adam optimization with default parameters [1,2], raising questions about the broader value of the proposed optimization strategy.


5. The authors' assertion that inadequate input representations are the primary challenge in PINN training overlooks other crucial factors identified in recent literature. Specifically, research has shown that derivative pathologies in deep MLP networks (independent of feature engineering) contribute significantly to optimization difficulties [1]. By focusing primarily on feature engineering, the paper may underestimate the importance of other architectural considerations, initialization schemes, and optimization challenges in PINN training. I doubt that feature engineering alone is the recipe to resolve the issues that PINNs face in challenging PDE settings.


6. The paper's discussion of feature map design based on dominant frequencies needs stronger validation on more challenging time-dependent problems. The Kuramoto-Sivashinsky equation (see comments above) would be a particularly valuable benchmark, as its solution exhibits time-varying dominant frequencies characteristic of chaotic systems. The current benchmarks all have relatively stable frequency characteristics, making it unclear how SAFE-NET's feature engineering strategies would adapt to problems where the relevant frequencies evolve significantly over time. Success with the KS equation would require either demonstrating that the proposed feature selection methods can capture temporally varying frequency structures, or acknowledging this limitation and proposing potential extensions. This benchmark would provide important insight into both the capabilities and limitations of the SAFE-NET framework for more complex time-dependent PDEs.


7. The paper's claims about SAFE-NET's novelty and differentiation from existing approaches in Section 3 are overstated. While the authors present SAFE-NET as significantly different from existing methods, similar ideas about engineered features in PINNs have been explored in previous work. For example, Delta-PINNs [3] and other approaches [4,5] have proposed tailored feature engineering strategies for specific PDEs. Rather than emphasizing novelty, the paper would be stronger if it positioned its contribution as a systematic exploration of an important open question: how to determine effective feature maps for different PDE systems. This reframing would better align with the paper's actual contributions while acknowledging its place in a broader line of research on feature engineering in PINNs.


[1] Wang, S., Li, B., Chen, Y., & Perdikaris, P. (2024). PirateNets: Physics-informed Deep Learning with Residual Adaptive Networks. arXiv preprint arXiv:2402.00326.

[2] Wang, S., Sankaran, S., Wang, H., & Perdikaris, P. (2023). An expert's guide to training physics-informed neural networks. arXiv preprint arXiv:2308.08468.

[3] Costabal, F. S., Pezzuto, S., & Perdikaris, P. (2024). Δ-PINNs: physics-informed neural networks on complex geometries. Engineering Applications of Artificial Intelligence, 127, 107324.

[4] Hu, W. F., Lin, T. S., & Lai, M. C. (2022). A discontinuity capturing shallow neural network for elliptic interface problems. Journal of Computational Physics, 469, 111576.

[5] Wu, J., Wang, S. F., & Perdikaris, P. (2023). A dive into spectral inference networks: improved algorithms for self-supervised learning of continuous spectral representations. Applied Mathematics and Mechanics, 44(7), 1199-1224.

**Questions:**

In addition to addressing the major concerns expressed above, please provide clarifications on the following questions:

1. Clarification is needed on the spectral density analysis, which is presented as a major contribution but lacks sufficient detail. Specifically:
   - What exact quantity does the spectral density represent in your analysis?
   - How is it computed in practice?
   - What is the mathematical connection between these spectral density plots and the conditioning of the PINNs loss landscape?
   - Figure 5's results require more rigorous interpretation to support claims about loss landscape regularization.

2. Given SAFE-NET's small parameter count (~600 parameters), the choice of optimization strategy warrants further explanation:
   - Why not use a second-order optimizer, which could be computationally feasible at this scale?
   - What is the theoretical or practical motivation for the specific sequence in (Adam + L-BFGS)²?
   - How does the computational cost compare to using a second-order optimizer?

3. The paper's approach to loss balancing requires clarification:
   - Was the decision to not use loss balancing deliberate?
   - Have you conducted experiments with loss balancing techniques?
   - How might loss balancing interact with your feature engineering strategies?
   - Could loss balancing further improve SAFE-NET's performance?

4. The evaluation metrics and reporting raise important questions, especially given the fact that the total training loss can often be a  misleading metric for assessing performance in PINNs:
   - Why rely primarily on PINN training loss when L2RE test error is often more informative?
   - Can you provide plots of individual loss components during training (residual loss, BC/IC losses)?
   - How do these components evolve differently with your feature engineering approach compared to baseline methods?
   - What is the relationship between training loss and actual solution quality in your experiments?

**Details Of Ethics Concerns:**

No ethics concerns to report.

---

### Official Review · Reviewer_S6kS · 2024-11-04

**Soundness:** 2
**Presentation:** 2
**Contribution:** 2
**Rating:** 5
**Confidence:** 4

**Summary:**

The authors propose a technique called SAFE-NET where the data augmentation for PINNs involves the combining the input data representation with hand-crafted features which helps with better conditioning for the PDEs and with the optimization and training process. The authors show an improvement in the epoch time as well as the number of epochs while matching or even exceeding the performance in some cases.
The work mainly suggests that with efficient feature engineering PINNs can utilize their representational capacity in a much more effective way even with much smaller and simpler models.

**Strengths:**

S1) The authors provide a parameter efficient (small model) approach for PINNs. Opposed to the larger model sizes in the traditional PINNs, the authors suggest a single-layer MLP network for PINNs

S2) A newer hybrid optimizer combining existing optimizers such as Adam and L-BFGS which converges faster and matches or exceeds baseline performance has been used.

S3) A major supporting factor and advantage for the experiments done with the new optimiser are the spectral density plots which visualize the effectiveness of the training procedure.

**Weaknesses:**

W1) There is no mention of the types of scenarios, equations etc. where the method fails (and where it works) such types of equations, regimes, initial or boundary conditions. The method does work on the canonical examples used in the experiments, however how does the feature engineering help with the learning is not clear.
Look at the question on the Logarithmic and Polynomial features (Q1)

W2) It has been mentioned multiple times that the network is a single layer network, however there is no clarity on the size of input, output and the hidden layer. There is a mention of the parameters being reduced from 5300 to 600, however it would be beneficial if all the sizes and corresponding layer wise parameters are mentioned clearly. (As this is one of the main contributions of the paper, reducing the model complexity and size)

W3) The proposed optimizer doesn’t have strong theoretical guarantees. See question below (Q3)

W4) This work doesn’t study complex high dimensional equations which are very standard in PINN/SciML literature such as Navier-Stokes (Kovasznay Flow etc.). What challenges they anticipate in applying SAFE-NET to such scenarios or scaling them?

**Presentation/Typos/Corrections**

Why is section 2 the Problem Setup and Section 3 as related work? (I believe this breaks the flow of the work and the sections could be interchanged). I also think the details about learning rates, discretization domains, mesh, points etc could have been included in the section 5 on experiments.

In section A.4 line 718, what is being referred by $K$ ?

In appendix A.2 Wave Equation, the parameter $\beta$ is not mentioned correctly, I believe the initial condition on line 682 should be $sin(\beta \pi x)$

In Appendix B, is there a subsequent B.2, B.3 section as the section is marked as B.1 ?

**Questions:**

Q1) In which specific scenarios do the logarithmic and polynomial features work? Why do they not work across all scenarios?

Q2) In the UIF (and others), where we uniformly sample n integers in the range $M_{min}$ to $M_{max}$ is this sampling with replacement or without replacement, that is can the same integer $k$ be picked more than once? (Same feature repeated, unsure if this helps the network or not)

Q3) How does the schedule of the proposed optimizer $(Adam+L-BFGS)^2$ look like? How many epochs of each optimizer are used ? Could the authors provide a general recipe which can be applied to other PINNs as well? Are there theoretical bounds and guarantees for the same?

---

### Official Review · Reviewer_XrGW · 2024-11-08

**Soundness:** 3
**Presentation:** 1
**Contribution:** 2
**Rating:** 3
**Confidence:** 2

**Summary:**

This paper presents SAFE-NET, a novel Physics-Informed Neural Network (PINN) architecture that leverages a single-layer structure combined with feature engineering, significantly reducing prediction errors and computational costs. SAFE-NET incorporates random Fourier features and an optimized learning approach, achieving high accuracy and fast convergence with fewer parameters compared to traditional multi-layer PINNs. It challenges the assumption that deep PINN architectures are effectively capturing relevant features in scientific applications. Numerical results demonstrate that SAFE-NET converges faster, enhances problem conditioning, and performs robustly across various problem settings, achieving comparable results with about 10 times fewer parameters in 20% of the epochs and 50% faster per epoch than conventional PINNs.

**Strengths:**

The authors introduce SAFE-NET, a model that achieves high efficiency by requiring fewer parameters than existing PINN models. It incorporates feature engineering through Fourier transforms, enabling a compact and powerful model design. By combining well-known optimizers, the model achieves faster training times and improved overall accuracy.

**Weaknesses:**

The paper lacks clarity in some explanations, making it challenging to fully understand key details. Specific areas needing improvement are:

**1. Data Processing**: The paper does not clearly explain how data is processed and fed into the model. It appears that Fourier transforms are used to focus on specific frequencies within the data, which may imply that SAFE-NET selectively captures frequency information. Could authors explain the exact feature engineering process, including any frequency selection criteria, and how the selected features are incorporated into the network architecture?. And could the authors clarify if the model indeed restricts itself to certain frequency bands, and if so, how this impacts performance and generalization?

**2. Hyperparameter Sensitivity**: The model appears to involve numerous hyperparameters, not only from the PINN setup but also from the feature engineering strategy. This complexity could suggest sensitivity to hyperparameter choices, yet the paper does not include ablation studies or analyses of hyperparameter robustness. Assessing how different parameter configurations impact model performance would provide a clearer understanding of SAFE-NET’s flexibility and reliability.

**3. Optimizer Explanation**: The use of Adam and L-BFGS optimizers, while promising, would benefit from further explanation. Though widely used, the rationale behind combining these optimizers specifically within SAFE-NET’s framework remains unclear. The paper should stand on its own in terms of clarity And The paper would be strengthened by a discussion of why neither optimizer alone suffices, as well as insights into how their combination uniquely contributes to performance improvements. This explanation would enhance the paper’s self-containment, allowing readers to appreciate the methodological choices without needing prior knowledge of these optimizers.

**Questions:**

Please refer to Weaknesses section.

---

### Note · Authors · 2024-11-15

I have read and agree with the venue's withdrawal policy on behalf of myself and my co-authors.